# Constitutional Mutation of *PIK3CA*: A Variant of Cowden Syndrome?

**DOI:** 10.3390/genes15091209

**Published:** 2024-09-15

**Authors:** Elena Vida-Navas, Verónica Barca-Tierno, Victoria López-Gómez, María Teresa Salazar, Miguel A. Moreno-Pelayo, Carmen Guillén-Ponce

**Affiliations:** 1Medical Oncology Department, Hospital Universitario Ramón y Cajal, Instituto Ramón y Cajal de Investigación Sanitaria (IRYCIS), 28034 Madrid, Spain; elenamaria.vida@salud.madrid.org (E.V.-N.); vlopezg@salud.madrid.org (V.L.-G.); mariateresa.salazar@salud.madrid.org (M.T.S.); 2Genetics Department, Hospital Universitario Ramón y Cajal, Instituto Ramón y Cajal de Investigación Sanitaria (IRYCIS), 28034 Madrid, Spain; veronica.barca@salud.madrid.org (V.B.-T.); mmorenop@salud.madrid.org (M.A.M.-P.); 3Centro de Investigación Biomédica en Red de Enfermedades Raras (CIBERER), 28034 Madrid, Spain

**Keywords:** *PIK3CA* germline mutation, Cowden syndrome, hereditary cancer, breast and ovarian hereditary cancer syndrome

## Abstract

We present a family in which four individuals have been identified with the same likely pathogenic genetic alteration in the *PIK3CA* gene at the germinal level; specifically, c.1145G>A p.(Arg382Lys) missense type. The index case patient was diagnosed with multinodular goiter and breast cancer at 61 years old. Among the other three carrier relatives: one has been diagnosed with serous cystadenoma of the ovary and a thyroid nodule with no radiological suspicion of malignancy; the other two present multinodular goiter. Additionally, a sister of three of the carriers suffered from an ovarian teratoma, follicular thyroid carcinoma on multinodular goiter, and high-grade serous ovarian carcinoma. No direct mutation study was performed on her as she had died due to ovarian carcinoma. This finding suggests that the *PIK3CA* gene should be considered in Cowden-like families when no other gene mutations have been found. Furthermore, this report contributes to characterization of the clinical phenotype caused by mutations in *PIK3CA*, which may be shared with other hereditary breast and ovarian cancer syndromes.

## 1. Introduction

Cowden syndrome (CS), also known as *PTEN* hamartoma tumor syndrome, increases the risk for benign and malignant tumors of the thyroid, breast, kidney, and endometrium. Consensus clinical diagnostic criteria for CS [1] and a scoring system based on the phenotype and age at diagnosis of CS [2] have been developed in order to identify *PTEN* germline mutations. When individuals with clinical features similar to CS do not meet the diagnostic criteria, they are referred to as CS-like (CSL) [3].

The majority of CS cases are a consequence of germline mutations in the *PTEN* gene, although other genetic and environmental factors may play a role in the differences in phenotypic expression. Moreover, only a minority of CLS individuals carry a *PTEN* mutation. The remaining non-*PTEN* genes must be associated with different molecular genetic mechanisms. It has been suggested also to consider a germline *KLLN* epimutation [4] and *SDHB-D* [5] analysis including *PIK3CA*, *AKT1* [6], *SEC23B* [7], and *WWP1* [8].

We describe a family history showing a possible association of *PIK3CA* gene mutation with breast and ovarian cancer and thyroid pathology, which adheres to phenotypic characteristics of a CLS.

## 2. Detailed Case Report

A 71-year-old woman with a past medical history of breast cancer was referred to our department. She was diagnosed in 2014 with grade 3 infiltrating ductal adenocarcinoma (pT1cN0M0), estrogen receptor positive (100%), progesterone receptor negative (0%) and HER2-negative. After that, she underwent annual mammography of the left breast, which showed a type C breast pattern, predominantly glandular, heterogeneously dense, with clusters of calcifications of benign morphology (BI-RADS 2). The patient has a history of hypothyroidism with a multinodular goiter, which was treated by right hemithyroidectomy in 1982 and had two punctures of a left thyroid nodule (2004 and 2009, respectively) which were negative for malignancy. The dermatological examination revealed a possible trichilemmoma on the dorsum of the nose and a few whitish papillomatous oral papules. The uterus was atrophic in the gynecological examination and, in 2023, a colonoscopy was performed with the finding of four subcentimetric polyps, two tubular adenomas with low-grade dysplasia, and another two hyperplastic polyps. Other antecedents include heavy smoking, grade 2 obesity (weight [W]: 87 kg (kg); height (h): 152 centimeters (cm); body mass index (BMI): 37.7 kg/m^2^) and chronic obstructive pulmonary disease grade of the Global Initiative for Chronic Obstructive Lung Disease (GOLD) 4B requiring chronic home oxygen. Her occipitofrontal circumference [OCF] measures 56 cm.

In the family, one sister suffered from an ovarian teratoma, a follicular thyroid carcinoma on multinodular goiter at age of 54 years, and she was diagnosed with a high-grade serous ovarian carcinoma at 57 years and died of metastases. Another sister (W: 67.9 kg, h: 153 cm, BMI: 29 kg/m^2^; OFC: 57 cm) had been diagnosed with serous cystadenoma of the ovary and a thyroid nodule with no radiological suspicion of malignancy; the other two presented multinodular goiter, and one of these (W: 78 kg) had a cutaneous dermatofibroma and a blue nevus. The proband had a healthy 38-year-old daughter (W: 76 kg, h: 163 cm, BMI: 28.6 kg/m^2^; OFC: 57 cm). The proband’s father died at 64 with no history of cancer and no siblings; while her mother is alive at the age of 94 with no history of malignancy, as are her five elderly siblings (Figure 1). In the proband, following informed consent, the molecular analysis was performed by using the Hereditary Plus OncokitDx V1 (CE-IVD, Imegen, San Diego, CA, USA). The principle of the assay relies on two key technologies: capture enrichment technology is used to amplify all coding sequence and exon-intron boundaries of the 50 genes associated with a predisposition towards cancer and massively parallel sequencing. These libraries were sequenced on the Nextseq platform (Illumina). The bioinformatics analysis was carried out through the Data Genomics (CE-IVD Imegen v.01) program. This software gives us information about SNPs (single nucleotide polymorphisms) and CNVs (copy number variations). All germline genetic alteration studies were performed on genomic DNA isolated from peripheral blood.

The molecular analysis includes all coding sequences and exon-intron boundaries of 50 genes involved in hereditary cancer and their location according to GRCh37: *APC (Chr5: 112,043,195-112,181,936), ATM (Chr11:108,093,211-108,239,829), BARD1 (Chr2:215,590,370-215,674,428), BMPR1A (Chr10: 88,516,407-88,692,595), BRCA1 (Chr17:41,196,312-41,277,500), BRCA2 (Chr13: 32,889,611-32,973,805), BRIP1 (Chr17:59,758,627-59,940,882), CDH1 (Chr16:68,771,128-68,869,451), CDK4 (chr12:58,141,510-58,149,796), CDKN2A (Chr9:21,967,751-21,995,300), CHEK2(Chr22:29,083,731-29,138,410), EPCAM (Chr2:47,572,297-47,614,740), FAM175A (Chr4: 84,382,092-84,406,334), FH (Chr1:241,660,903-241,683,061), KIF1B (Chr1: 10,270,863-10,441,661), MAX (Chr14: 65,472,892-65,569,413), MEN1 (Chr11: 64,570,982-64,578,766), MET (Chr7: 116,312,444-116,438,440), MLH1 (Chr3:37,034,823-37,107,380), MLH3 (Chr14: 75,480,467-75,518,235), MRE11A (Chr11: 94,152,895-94,227,074), MSH2 (Chr2: 47,630,108-47,789,450), MSH3 (Chr5:79,950,467-80,172,279), MSH6 (Chr2: 47,922,669-48,037,240), MUTYH (Chr1:45,794,835-45,806,142), NBN (Chr8: 90,945,564-91,015,456), NF1 (Chr17:29,421,945-29,709,134), NTHL1 (Chr16: 2,089,816-2,097,867), PALB2 (Chr16:23,614,488-23,652,631), PIK3CA (Chr3: 178,865,902-178,957,881), PMS2 (Chr7:6,012,870-6,048,756), POLD1 (Chr19: 50,887,461-50,921,273), POLE (Chr12:133,200,348-133,263,951), PTEN (Chr10: 89,622,870-89,731,687), RAD50 (Chr5: 131,891,711-131,980,313), RAD51C (Chr17: 56,769,934-56,811,703), RAD51D (Chr17:33,426,811-33,448,541), RB1 Chr(13: 48,877,887-49,056,122), RET (Chr10: 43,572,475-43,625,799), SDHA (Chr218,356-256,815), SDHAF2 (Chr11:61,197,514-61,215,001), SDHB (Chr17,345,217-17,380,665), SDHC (Chr1:161,284,047-161,332,984), SDHD (Chr111,957,497-111,990,353), SMAD4 (Chr18:48,494,410-48,611,415), STK11 (Chr1,189,406-1,228,428), TMEM127 (Chr2:96,914,254-96,931,732), TP53 (Chr7,565,097-7,590,856), VHL (Chr3: 10,182,692-10,193,904),* and *XRCC2 (Chr7:152,341,864-152,373,250).*

In the rest of the family members, the analysis of likely pathogenic variants detected by NGS was confirmed by polymerase chain reaction (PCR) amplification and direct sequencing.

The study identified in heterozygosis a likely pathogenic variant (class IV) according to the ACMG guidelines [9] for the classification of sequence variants (PP3 strong, PM1, PM2, and PP5 supporting): NM_006218.4: c.1145G>A p.(Arg382Lys) located in exon 6 of the *PIK3CA* gene (Figure 2). The mutation identified in the *PIK3CA* gene was also confirmed in both living sisters and the proband’s daughter (marked with a dot in Figure 1).

## 3. Discussion

CS, also known as *PTEN* hamartoma tumor syndrome, is an autosomal dominant hereditary disorder that increases the risk of developing cancer and hamartomatous lesions in various tissues, including the gastrointestinal tract, skin, mucous membranes, breast, thyroid, endometrium, and nervous system. The lifetime risk of developing some form of cancer is as high as 89%, with the most common types being breast cancer in women, thyroid cancer, endometrial cancer, and renal cancer [10].

The diagnosis of CS is initially clinical, requiring the fulfillment of a set of established criteria [1], and is confirmed through genetic testing, with pathogenic variants of the *PTEN* gene being the most common. Although it was previously described that 85% of patients meeting the diagnostic criteria carry *PTEN* mutations, recent findings suggest that the actual figure is closer to 30–35% [11]. There are other less common mutations in genes such as *SDHx* [5] or germline hypermethylation of the *KLLN* gene [4]. However, some patients with clinical criteria for Cowden syndrome do not have any of the described mutations associated with the syndrome.

In a study including 91 patients with clinical manifestations compatible with Cowden syndrome but without constitutional *PTEN*, *SDHx* mutations, or *KLLN* hypermethylation, referred to as CLS, germline mutations in the *PIK3CA* gene were found in eight individuals and two in the *AKT* gene. Specifically, they described two *PIK3CA* mutations that could be implicated in the development of this syndrome: c.353G>A and c.1145G>A, resulting in the proteins p.Gly118Asp and p.Arg382Lys. These mutations affect the C2 domain of the PIK3CA protein, which is responsible for the recruitment of p110α to the cell membrane. This study revealed increased levels of PIP3 in PIK3CA-p.Glu218Lys and moderately increased levels in PIK3CA-p.Arg382Lys cells compared to the wild type (WT) [6]. The other variants identified affect different domains of the protein, such as PIK3-RBD or PIK-Helical.

The *PTEN* gene is a tumor suppressor gene that transcribes the *PTEN* protein, which negatively regulates the phosphatidylinositol-3-kinase (PI3K) cell signaling pathway. The final effect of the PI3K pathway activation promotes cell growth, proliferation, and survival [12]. The close molecular relationship between *PTEN* and *PIK3CA* explains similar clinical manifestations in mutations of these genes, with tumor development in the breast or thyroid, as described.

We describe a new family with at least four individuals carrying one of the described mutations, the c.1145G>A variant of the *PIK3CA* gene, with clinical manifestations compatible with the Cowden syndrome spectrum. All patients present multinodular goiter, with family cases of breast cancer and follicular thyroid carcinoma. This supports the inclusion of *PIK3CA* variants as a cause of the previously described Cowden-like syndrome.

However, due to the rarity of this finding, the penetrance of this mutation and the spectrum of associated tumors are not clear. A possible reason is that the likely pathogenic variant identified in our patient could be a hypomorphic variant. This type of variant decreases, but does not eliminate, the gene function. Therefore, not all family members developed the pathology. Thus, although the follow-up of individuals with *PTEN* mutations is well established, with clinical guidelines available [13], many questions remain regarding the most appropriate screening in individuals with Cowden-like syndrome carrying other mutations in genes such as *AKT* and *PIK3CA*. There is insufficient evidence in the literature on this, with this being the ninth family identified with a pathogenic germline variant of this gene.

The low frequency of pathogenic variants of this gene in the germline could be explained by the development of fetal malformations in the offspring, such as fetal overgrowth or macrocephaly, often incompatible with life, which hinders the hereditary transmission of these variants [14,15]. In the family we describe, there have been no cases of miscarriages, macrocephaly, or fetal malformations, so other mutations are likely responsible for these alterations.

In conclusion, although there is still little evidence, constitutional pathogenic variants of *PIK3CA* could be the cause of Cowden-like syndrome and should therefore be considered in the genetic diagnosis of these patients. Constitutional mutations of the *PIK3CA* gene appear to increase the risk of breast cancer, gynecologic tumors, and thyroid pathology. There is not enough evidence to establish follow-up recommendations for these patients.

## Figures and Tables

**Figure 1 genes-15-01209-f001:**
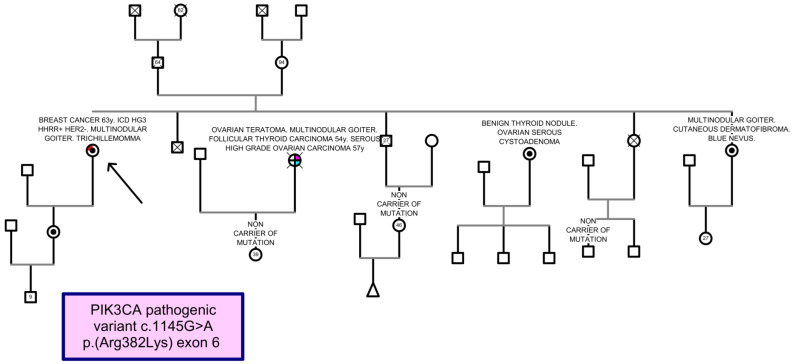
Pedigree of family studied. The black arrow indicates the index case studied. The variant has been identified in four members (black plot). Cancer diagnosis is indicated as follows: red corresponds to breast cancer, pink corresponds to ovarian cancer, and blue corresponds to thyroid cancer. Crossed individuals are deceased.

**Figure 2 genes-15-01209-f002:**
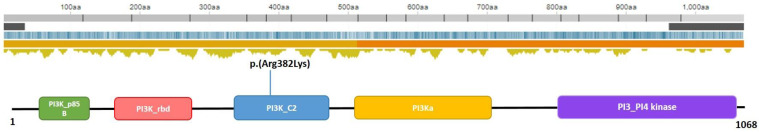
Schematic of the PIK3CA protein, depicted with its functional domains. The likely pathogenic 124 variant is located motif within the PI3K_C2, responsible for the recruitment of p110α to the cell membrane, at arginine 382 (350–485aa).

## Data Availability

The original contributions presented in the study are included in the article, further inquiries can be directed to the corresponding author.

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
