# Peer review of "Constitutional Mutation of PIK3CA: A Variant of Cowden Syndrome?"

_genes, 2024, doi:10.3390/genes15091209_

Round 1

Reviewer 1 Report

Comments and Suggestions for Authors

This manuscript does add some additional evidence that PIK3CA PVs are associated with a Cowden like phenotype.  Would suggest the following

1. cut the length down.  Don't need all the detail about the patients breast cancer treatment. Some of her past history is extraneous.

2. Figure 1 needs better clarity with respect to color coding of cases in legend.  I don't see an arrow for index case in my manuscript

3. Would benefit from a native english speaker polishing up grammar

4.  The online calculator is used to determine likelyhood of a PTEN mutation.  Not described CS/CLS

5.  Genes other than PTEN have been implicated in a Cowden like syndrome, not, to my knowledge Cowden syndrome itself.

6. Take out the paragraph about PIK3CA mutations causing in -utero death as a means of them being infrequent.  Not relevant here -- and highlights a big weakness of the MS.  Do any of your cases have macrocephaly???  If you did not examine them for macrocephaly or it is absent you do need to state this. 

Comments on the Quality of English Language

See above.  

Author Response

Thank you very much for the review that have been carried out. The changes proposed have allowed us to significantly improve our manuscript and make it more understandable.

Comments 1: cut the length down.  Don't need all the detail about the patients breast cancer treatment. Some of her past history is extraneous.

Response 1: Thank you for pointing this out. We agree with this comment. Therefore, we have reduced the cancer history of the patients, focusing on the diagnosis and limiting the information about treatments. Changes can be found in page 2 paragraph 1 lines 45-50.

Comments 2. Figure 1 needs better clarity with respect to color coding of cases in legend.  I don't see an arrow for index case in my manuscript.

Response 2: Thank you for pointing this out. We agree with this comment. Therefore, we have modified the figure including the arrow, which has been disappeared. We have also added the color coding to clarify the diagnosis of the patients. Changes can be found on Figure 1 and Legend of Figure 1 (in page 3 paragraph 1 lines 80-82).

Comments 3: Would benefit from a native english speaker polishing up grammar

Response 3: Thank you for pointing this out. We have reviewed the English grammar in the full document and made minor changes.

Comments 4.  The online calculator is used to determine likelyhood of a PTEN mutation.  Not described CS/CLS.

Response 4: Thank you for pointing this out. Our text may be confusing, we have modified ir in order to explain betther this concept. Changes can be found in page 1 paragraph 1 lines 30-33.

Comments 5: Genes other than PTEN have been implicated in a Cowden like syndrome, not, to my knowledge Cowden syndrome itself.

Response 5: Thank you for pointing this out. As you can see in the bibliography that we have added to the manuscript, other genes have been identified in CS patients, like SDHx or KLLN (references 4 and 5) or PIK3CA and AKT (reference 6).

Comments 6: Take out the paragraph about PIK3CA mutations causing in -utero death as a means of them being infrequent.  Not relevant here -- and highlights a big weakness of the MS.  Do any of your cases have macrocephaly???  If you did not examine them for macrocephaly or it is absent you do need to state this. 

Response 6: Thank you for pointing this out. We have tried to remark the infrequency of this finding in order to explain the lack of evidence about the management and identification of these patients, and the need of further research. None of our patients have macrocephaly and we have included it in the reviewed manuscript (on page 4 paragraph 4 lines 153-155).

Reviewer 2 Report

Comments and Suggestions for Authors

Although there is just little evidence, this review can still provide some clues for the relationship of PIK3CA variants and the cause of Cowden-like syndrome. Here are some comments:

1. In line 111, the sentence “other less common mutations in such as SDHx or…KLLN gene” lacks references.

2. In line 142, the authors wrote that “…this being the ninth family…”, how about the variant site in the other eight families?

3.In figure 1, the black arrow is not visible, so replace it with one that is clearly distinguishable from the background.

4.In figure 1, please mark the meaning of the “×” and the red/blue color in the figure.

Author Response

Thank you very much for the review. The changes proposed have allowed us to significantly improve our manuscript and make it more understandable.

Comments 1: In line 111, the sentence “other less common mutations in such as SDHx or…KLLN gene” lacks references.

Response 1: Thank you for pointing this out. We agree with you and we have added the references in the new versión. Changes can be found on page 1 paragraph 2 lines 39-40 and on page 3 paragraph 7 lines 113 and 116, and page 3 paragraph 8 lines 117-120.

Comments 2: In line 142, the authors wrote that “…this being the ninth family…”, how about the variant site in the other eight families?

Response 2: Thank you for pointing this out. We have mentioned them on page 3 paragraph 8 lines 117-127, but after consider you suggestion we have explained more about them on this part of the discussion. Changes can be found on pages 3-4 lines 125-127.

Comments 3: In figure 1, the black arrow is not visible, so replace it with one that is clearly distinguishable from the background.

Response 3: Thank you for pointing this out. We agree with you and have added again the black arrow in order to make it more visible. Change can be found on Figure 1.

Comment 4: In figure 1, please mark the meaning of the “×” and the red/blue color in the figure.

Response 4: Thank you for pointing this out. We agree with this comment. Therefore, we have added the color coding to clarify the diagnosis of the patients. Changes can be found on and Legend of Figure 1 (in page 3 paragraph 1 lines 80-82).

In addition, we have improved the description of the methodology and results of genetic studies.

Finally, we have modified our conclusions in order to include all our findings (in page 4 paragraph 5 lines 156-158).

Round 2

Reviewer 2 Report

Comments and Suggestions for Authors

This manuscript has been sufficiently improved to warrant publication in Genes.

Author Response

Thank you very much for your review.

We greatly appreciate your comments, which have undoubtedly improved the manuscript for publication.